# Optical Methods of Error Detection in Additive Manufacturing: A Literature Review

**Brianna Wylie * and Carl Moore Jr.**

Department of Mechanical Engineering, Florida A&M University, CISCOR Lab,
FAMU-FSU College of Engineering, Tallahassee, FL 32310, USA
* Correspondence: brianna.wylie@famu.edu; Tel.: +1-813-316-5784

**Abstract:** Additive Manufacturing (AM) has been a growing industry, specifically when trying to mass produce products more cheaply and efficiently. However, there are too many current setbacks for AM to replace traditional production methods. One of the major problems with 3D printing is the high error rate compared to other forms of production. These high error rates lead to wasted material and valuable time. Furthermore, even when parts do not result in total failure, the outcome can often be less than desirable, with minor misprints or porosity causing weaknesses in the product. To help mitigate error and better understand the quality of a given print, the field of AM monitoring in research has been ever-growing. This paper looks through the literature on two AM processes: fused deposition modeling (FDM) and laser bed powder fusion (LBPF) printers, to see the current process monitoring architecture. The review focuses on the optical monitoring of 3D printing and separates the studies by type of camera. This review then summarizes specific trends in literature, points out the current limitations of the field of research, and finally suggests architecture and research focuses that will help forward the process monitoring field.

**Keywords:** additive manufacturing; 3D Printing; artificial intelligence; optical monitoring; literature review

## 1. Introduction

Additive manufacturing (AM) is one of the fastest-growing industries in our modern tech world. In particular, parts with complex internal geometries can be built in a single unit, simplifying productions that traditionally required multi-step manufacturing methods [1]. AM's printing process typically stacks material layer-by-layer to construct three-dimensional products based on 3D CAD models [2]. Due to the advantages of low production lead time and the ability to create complicated geometries and shapes, the process has uses in various industrial applications [3]. AM witnessed its first development in the mid-1980s, where it was solely capable of processing polymers using technologies such as stereolithography (SLA) [4]. Since then, this manufacturing process has produced multiple disciplines and categories. There are generally seven AM categories: Photopolymerization, extrusion, sheet lamination, beam deposition, direct write and printing, powder bed, binder jet printing, and powder bed fusion [1]. The main types of AM this review will focus on will be extrusion and powder bed fusion, as they are some of the most prevalent types of AM used in the industry [5].

One key barrier that prevents extrusion-based AM from being applied to more industrial applications is the relatively low fabrication quality in dimensional accuracy and mechanical strength [6]. For instance, fused filament fabrication (FFF) has significantly lower reliability levels than other manufacturing processes, with research estimating 20% printing failure rate by unskilled users. A process that exists to monitor a print, notice when an error is occurring, and know what parameters to fix in the system to correct the mistake would be cost-effective in money and time. Achieving high levels of quality and repeatability of AM parts is a highly challenging task due to many factors. Such factors

include the complexity of the underlying physical phenomena and transformations during part production and the lack of formal mathematical and statistical models needed to control the build process and ensure the quality of the parts [4]. Based on the need for better quality products produced by AM processes, the field of AM process monitoring developed. The ultimate goal of additive manufacturing process monitoring is to create effective real-time, closed-loop feedback control of the additive process [1].

While relatively young (only really taking off in the past two decades), the field of research monitoring and controlling AM manufacturing has branched off into many areas. First and foremost, the areas branch off depending on the type of AM manufacturing process. As stated before, the focus of this paper will be on FFF, a.k.a. FDM printing and laser bed fusion printing (LBFP), as these are some of the most common forms of AM manufacturing and have the most extensive wealth of material on AM monitoring.

FFF is an AM process in which a workpiece is manufactured by depositing progressive layers of extruded molten material. Among AM technologies, the most widespread process is Fused Deposition Modeling (FDM), patented initially by Stratasys company [7]. In FFF, a thermoplastic material is typically heated past its glass transition temperature and extruded through a nozzle in a controlled manner [8]. For this type of print, there are generally two ways to monitor AM processes: monitoring the printer's health state and detecting product defects. In terms of tracking the printer, the significant places to observe for FFF are first the hot end(the part of the extruder that heats the thermoplastics) and the second most prevalent monitoring spot is the cold end (the part that feeds the thermoplastics to the hot end) [3]. See Figure 1 to see a diagram of the extrusion drive of an FDM printer [9].

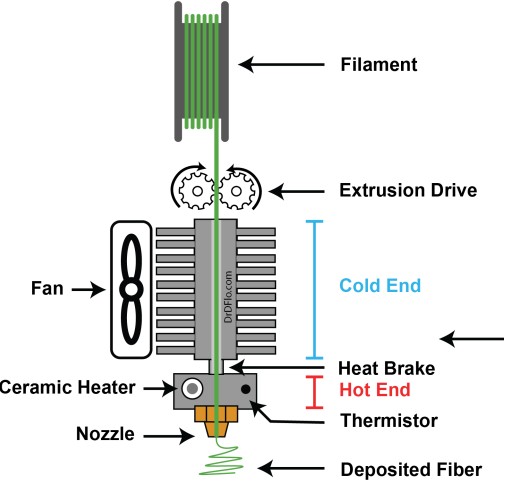

**Figure 1.** The extrusion drive pushes filament into the extruder, which liquifies the plastic. The molten plastic is deposited as a fiber that is approximately the diameter of the nozzle. The cold end and heat break localize the heat to the hot end [9].

Laser PBF (L-PBF) machines (also known as selective laser melting (SLM) machines) spread a thin layer of metal powder, typically 20–120 μm thick, across a substrate often referred to as a "build plate". After spreading the powder layer, one or more laser beams are used to selectively melt the powder in the shape of a 2D slice of a 3D part. After the lasing is complete, the build plate lowers, another layer of powder deposits on the powder bed, and the process repeats until the end of print [10]. Defects and flaws in powder-bed AM appear within the layer. A common and critical form of defect is the formation of pores in part. Porosity can reduce the part density and the structural and mechanical properties such as tensile strength and fatigue properties [11]. Research generally categorizes LPBF monitoring methods into two main groups: built surface monitoring and melt pool monitoring. The melt pool monitoring has attracted much attention as the melt pool status directly determines built quality [12].

To accurately correct and locate error detection, understanding the underlying mechanical properties and how changes in the mechanical properties affect the print is necessary.

One of the major parameter structures is the build orientation. Build orientation refers to how and in which direction a sample is placed on the 3D printing platform. This is often observed in the form of anisotropically printed objects, making structural performance highly dependent on build orientation similarly to composite laminates [13]. Hambali et al. noticed that in physical testing of FDM parts, when pulling direction is parallel to build direction, the parts are stronger but they tend to crack in a specific place where shear occured. They also found that different oriented parts can show different deformation behaviours so orientation direction of the parts should be chosen according to boundary conditions [14]. Harris et al. found that inter-layer fusion and trans-layer failure was greatly affected by build orientation. Inter-layer fusion is the bond of the lower layer with the one currently being extruded and the trans-layer fusion bond is the fusion roads of the same layer [15]. In vertical samples, inter-layer fusion bond broke due to the applied load parallel to the extruded layer.

The deposited layer bore the whole force instead of individual beads leading to low strength causing interlayer failure. While the load applied perpendicular to the deposited layer, in flat and on-edge samples, made the beads bear the applied load, resulting in high strength. The literature consensus shows that the most robust printing orientation is when fused filament deposition coincides with the pull direction. However, a range of orientations may be found along this pull direction [13].

Feed rate also known as scanning speed is the speed of the nozzle motion. This can be divided into two profiles of scanning speed and filling speed. Directly tied to filament speed is extrusion speed that refers to the speed of filament extrusion from nozzle [16]. Lower scanning speed causes processing inefficiencies, the layer processed will be burned and destroyed by the hot nozzle. Higher speed produces mechanical vibrations, detrimental to parts accuracy and if scanning speed is much greater than extrusion speed, filament will be pulled to be too fine. In terms of material properties, an increase in feed rate shows a decrease in tensile and flexural strengths [17]. Feed rate is also directly related to build time and therefore manufacturing cost [5].

Layer thickness seems to show a disparity in results as it seems coupled to other process parameters. For instance in a study by Chacón et al. [13], with build orientation, in upright samples, tensile and flexural strengths incereased as layer thickness increased. Where layer thickness had little effect on flat or on-edge orientations.

Other things of note with process parameters include that structural parameters have a greater influence on the mechanical properties of FFF components than manufacturing parameters. The Ishikawa Diagram in Figure 2 shows examples of what these parameters would be [5]. Also, the material in the FDM printer greatly affects the material properties and thus studies on how the process paramaters affect each individual FDM material have been undergone [15,18,19].

As far as take away points to understand when monitoring part faults a couple of observations were found [5]:

- Design of Experiments(DOE) is useful for multi-parameter studies. However, individual contributions of parameters are hidden because of the number of variables and unknowns that are tied to the process
- A need for uniform testing standard for AM manufacturing is evident by reviewing the experimental studies done on the matter
- For modelling real prints, it is beneficial to characterize the process and build parameters along the boundary conditions that best describes the heterogeneity of the printed part
- Little research has been done on mechanical properties of FFF parts with low densities with different loadings

- Low porosity static samples resulted in highest mechanical properties. Meanwhile, dynamic properties had improved damping behavior when increasing porosity in a print
- Properties that are sensitive to processing parameters like inter-layer bonding should be included to accurately model print failure.

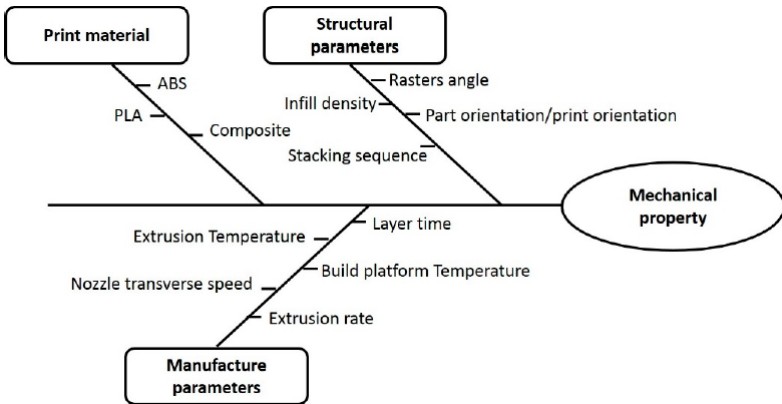

**Figure 2.** Ishikawa Diagram showing the different causes of different mechanical properties in a print [5].

LPBF studies investigating how process parameters affect the print have also been reviewed. A study done by Galy et al. [20] showed that the principle defects of Al alloyed parts are porosity, hot cracking, anisotropy, and surface quality. They also show of that the specific SLM process parameters as well as laser beam energy loss due to Al reflectivity are the main causes of porosity and hot crack formation. Meanwhile Kleszczynski et al. [21] cited typical process errors and mapped them to possible causes seen in Figure 3.

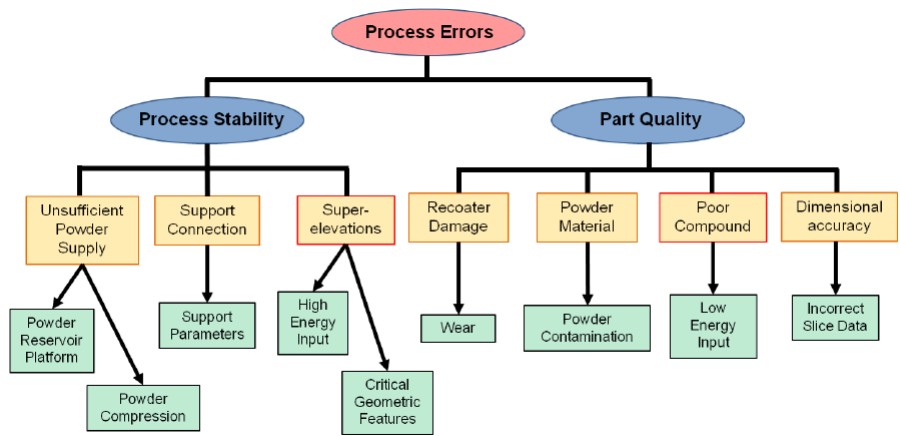

**Figure 3.** Typical Process Errors in Laser Beam Melting categorized by influence, type and cause [21].

In terms of process parameters for LPBF most of the controllable process parameters are tied together by a function called energy density [22]. The energy density function is defined by the equation below where $E$ is the laser energy density, $P$ is the laser power, $S$ is the scanning speed, $T$ is the layer thickness, and $H$ is the hatch distance [23].

$$E = \frac{P}{S \times T \times H} \tag{1}$$

Some of the conclusion that can be drawn from the process parameter of LBPF printing include:

- Al alloys have a spherical partical shape mixed with a large percentage of elongated particle

- Energy density largely affected relative density and the formation of pores ins prints. High energy density rates contribute to large hydrogen spherical pores forming, while lower rates created keyhole pores
- Scan speed and laser power have a close relationships where the highest relative density was achieved at low scan speed and power and lower density values
- Surface topology was signficantly affected by energy density
- Best surface flatness could be obtained with higher hatch spacing and scan speed for certain Al alloys
- For certain Al alloys, contraction of part dimension was observed at lower energy densities and oversized part dimension was detected at higher energy densities.

There have been a fair amount of literature reviews on AM monitoring, generally focusing on a specific type of printer or error. However, there has been little focus on specific setups of monitoring systems, reviews on which designs work for certain printers, or specific errors trying to be detected. Creating a study that shows which architectures work for a given error detection will enable researchers to find the particular area of AM process monitoring that they want to improve upon quicker and quickly jump into creating their setup. Researchers use multiple sensors to monitor AM processes; however, this literature review will focus primarily on the optical sensor types. The review covers the kind of optical detection used to detect errors. Many of the studies reviewed also have multi-sensor setups. Within each area, the systems will differentiate the printer being used (LBFP or FFF) and the errors that the architecture focuses on detecting. After a quick review of the different optical methods, the study will discuss trends on what architectures are helpful for given errors. Finally, the study will mention the given error and detection architecture intended to be used by the author's research.

## 2. Camera Setups

This section goes through the most commonly used cameras in AM optical monitoring. The unit will briefly describe what the camera does and what specific errors and uses the camera tackles.

### 2.1. CCD (Charge Couple Device) Camera

A CCD camera is a solid-state electrical device that converts light input into an electronic signal. The term charged-coupled refers to the coupling of electrical potentials within the silicon material's chemical structure that comprises the chip's layers [24]. The main advantages of state-of-the-art CCD detectors and digital imaging are high time resolutions and a good linear dynamic range of up to 16 bits, which ensures high accuracy for signal variations [25]. In Figure 4 below you can see a schematic of a CCD camera.

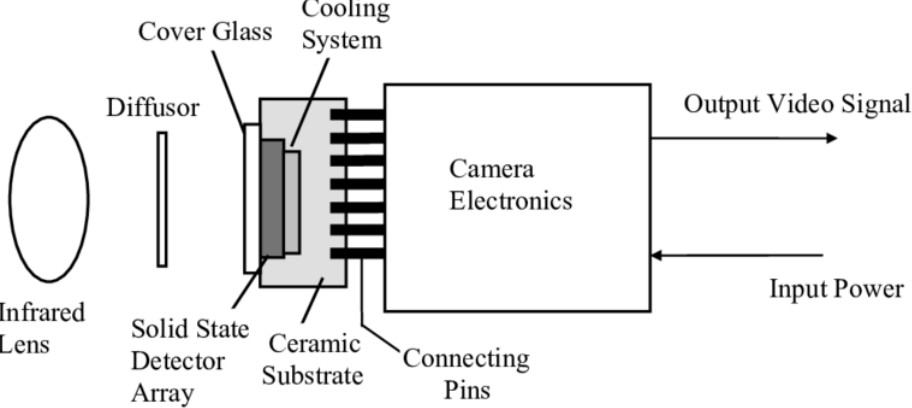

**Figure 4.** Schematic of CCD camera [26].

Optical monitoring of LBPF printers often utilizes CCD cameras. In a study by Kleszczynski et al., the team creates an optical architecture to detect irregularities in the print. The camera was mounted in front of the printer window and captured images of the build platform from an observation angle. A tilt and shift lens helped to reduce perspective distortion by shifting the camera back and allowed placing the focal plane on the build platform without stopping down using its tilt ability [21]. Figure 5 shows the architecture of the LPBF printer and camera.

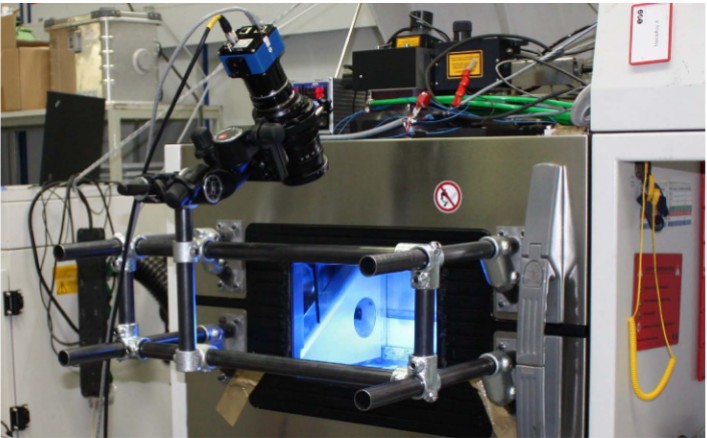

**Figure 5.** Architecture of Kleszczynki et al. [27].

The team created prints with severe overhang without support and other geometries they knew would be prone to failure and manually documented and processed common process errors (see Figure 3). Different scanning methods that depended on changing scanning speed were utilized for the geometries that studied support structures to see the effects. Meanwhile, for overhang, the different overhanging angles were tested with different process parameters as a critical overhanging angle was prominent no matter what strategy was used to print. In their other works, they used their optical architecture to detect super-elevation [27,28]. Super-elevation occurs when a part warps or curls upward out of the powder layer, typically a result of buildup residual thermal stresses [29]. Super-elevation of parts during the LBM fabrication poses a large problem to process stability as collisions between coater mechanism and part can lead to damage of both part and machine. They then created a way to auto-segment their images taken [30].

Zeinali and Khaajepour [31] developed a closed-loop control system for an SLM printer using a CCD camera to monitor the height of any printer layer. The study created a control law based on the following model.

$$\tau\dot{h} + h = \frac{3}{2}\frac{\dot{m}}{\rho w_0}\left(\frac{d_b}{d_p}\left(1 + \frac{h}{d_b}\tan\alpha\right)\cos\alpha\right)^n \frac{k}{v(t)} \tag{2}$$

where $h$ is the clad height in mm, $\dot{h}$ is the derivative of the clad height, $\dot{m}$ is the powder flow rate, $\rho$ is the powder density, $w_0$ is the steady-state values, $d_p$ is jet diameter, $d_b$ is the laser beam diameter, $\alpha$ is the angle between nozzle and laser beam, $v(t)$ is scanning speed, and $\tau$, $v$, and k are unknown parameters that are identified experimentally offline. Therefore the data from the CCD camera helps inform the model of the printer and thus better tunes the closed-loop control. The image processing algorithm (IMPA) was installed on the CCD camera to attenuate high intensity and eliminate flare and noise. See the experimental setup in Figure 6. The adaptive sliding mode of control contributed to process monitoring by eliminating chattering (high-frequency oscillations) and being able to estimate lumped uncertainty as an uncertain part of the dynamic model instead of having conservative estimated uncertainty bounds.

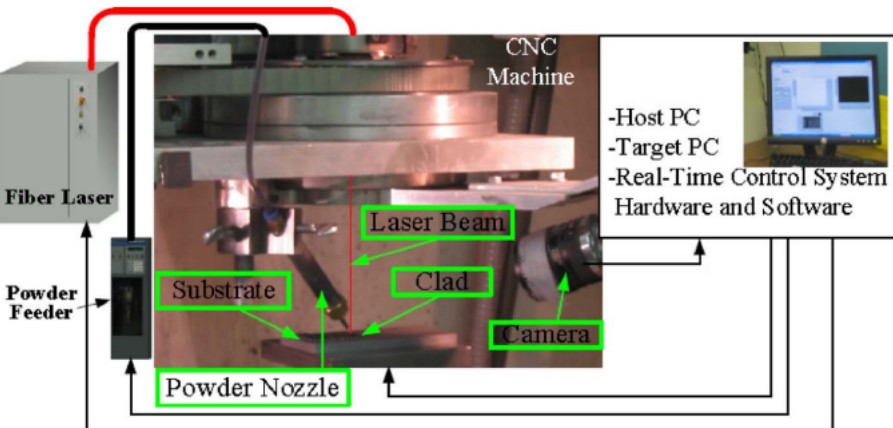

**Figure 6.** Laser Cladding Setup [31].

Although not technically FDM fused deposition of ceramics (FDC) is an extension of FDM and has a similar style of operation. A group of studies by Cheng and Jafari used a CCD camera to minimize assignable defects in print [32]. The study built on the work of a study done in 1998 that also used FDC but utilized a stereotype camera setup instead [33]. Assignable defects have defect patterns linked to a given cause. An example of such a defect is inappropriate parameter settings. These defects can then be eliminated or reduced with control parameters. The group applied a 2-D profiling algorithm to find representative figures of both underfill and overfill. Signatures, a simple representation of an object or a process in the form of a mathematical function, a feature vector, a geometric shape, or some other model, were used to create the representative figures. Figure 7 shows a visual of the experimental setup [34].

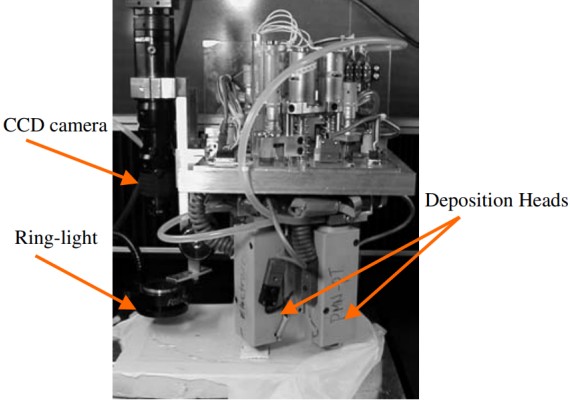

**Figure 7.** Machine Vision system of FDC machine [34].

In layered manufacturing applications, a single layer is a set of a boundary and several interior regions. Therefore, the defect detection problem breaks down into two issues: partitioning the image into regularities or homogeneous regions and detecting abnormal or unexpected signatures, which could indicate defects. Using the signature for each uniform part, an image, which is called the ideal image, can be reconstructed. Detecting defective areas is done by comparing the original and the perfect image. An FDC image processing and defect detection software package, FipSoft 1.0, has been developed using Visual C++ on the Window NT 4.0 platform to perform online image processing and defect detection software packages. It provides a friendly user interface and many modules to implement the signature analysis methodology and integrate the machine vision system with the FDC control system. The system spots the error, classifies underfill or overfill based on surface texture, validates the data, and creates a control model. The road width is affected by many

process variables, including flow rate, roller speed, head speed, temperature, and distance between the head and part surface. In their specific control model, roller speed was the process parameter used as the control variable.

Shen et al. changed the end effector of a MITSUBISHI six degrees of freedom (DOF) robot to include a nozzle, heat block, cooling fan, photoelectric switch, filament extruding motor, CCD camera and light system to capture in-situ images of a print bed [35]. Their setup can be seen in Figures 8 and 9.

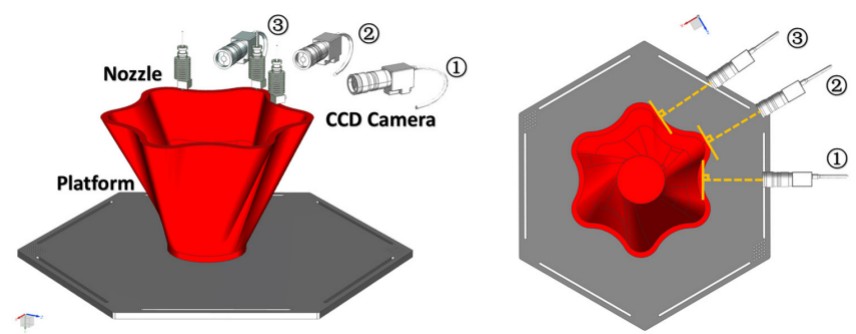

**Figure 8.** Principle of multi-view vision detection system [35]. The numbers represent the different camera angles the CCD camera is positioned.

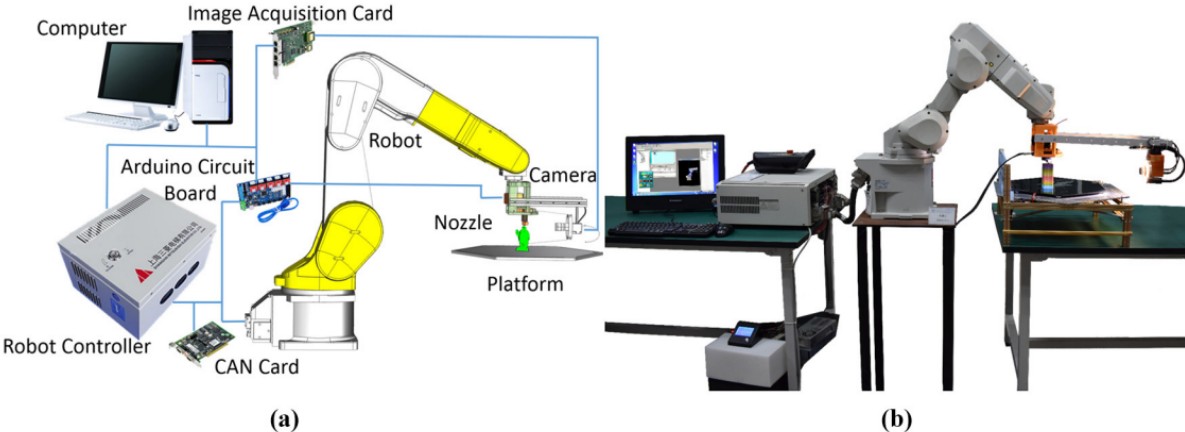

**Figure 9.** Hardware system structures of robot FDM system (**a**) Control components (**b**) Physical Platform [35].

A vision detection system was created based on Visual C++ Integrated Development Environment and Open Source Computer Vision Library. The team roughly classified defects into three typical types according to aspect ratio and area distribution. The three defects are transverse defects, longitudinal defects, and localized defects. A transverse defect is when the aspect ratio of the major defect is larger than 3. This defect occurs when their is either excessive or insufficient extrusion. This causes a mismatch of motion. A longitudinal defect occurs when their is a low aspect ratio that is generally less than 0.4. Longitudinal defects are mainly caused by the mechanical structure and mechanical motion. The other defects are defined as localized defects, including defects with scattered area distribution. A table showing the conclusions of defects found and possible causes are seen in Table 1.

**Table 1.** Possible causes of three typical types of defects [35].

| Defect Type | Judgement | Some Possible Reasons |
|---|---|---|
| Transverse defect | Excessive estrusion | The temperature of the nozzle is too high; Static flow beading and dynamic flow beading |
| | Insufficient extrusion | The nozzle is blocked; The temperature of the nozzle is too low; The filament is used up; The extruding motor does not work; The thickness of the layer is too large |
| | Mismatch between motion and extrusion | The communication failure between Atmega2560 and robot controller; The velocity of movement is too fast |
| Longitudinal defect | Mechanical structure | It is located in the turning point of the robot motion |
| | Mechanical motion | The acceleration at the start or end point of each layer is too small |
| Localized defect | Unreasonable temperature | It can't cool quickly for excessive melting; The temperature fluctuaion |
| | Mechanical problem | The excessive vibration caused by loosening and so on |

Multi-Sensor CCD Camera Systems

Many studies also employed a multi-sensor approach to monitoring, employing additional sensors with the CCD camera to better understand what was happening during the printing session. A study by Doubenskaia uses a pyrometer and a CCD camera to monitor thermal processes in SLM and compares the results obtained by different diagnostic tools [36]. The optical architecture can be seen in Figure 10. The temperature was measured in the laser impact zone by a bi-color pyrometer. In contrast, the CCD camera acquired the brightness temperature distribution in the heat-affected zone (HAZ). In the study, researchers analyze variations in pyrometer signal and brightness temperature with operational parameters by variations in hatch distance and powder layer thickness. Hatch distance is the distance between the neighbor tracks. A track is a feature that results from the laser beam scanning along a straight line on the powder bed with constant speed. The paper instead of monitoring a specific print error looked at how the process parameters that were controlled had effects on each other and also how they affected the temperature distribution of the print.

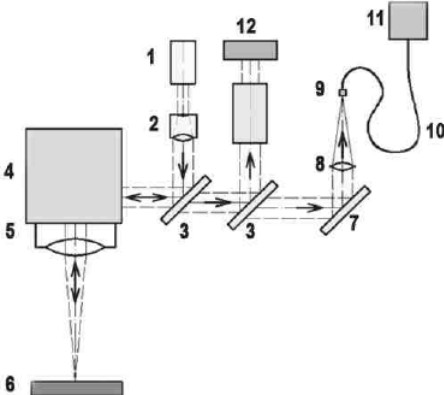

**Figure 10.** Schematic of the optical system applied: 1—fiber laser; 2—beam expander; 3—laser beam/thermal signal separating mirror; 4—scanner head; 5—F-Theta lens; 6—powder bed; 7—mirror; 8—pyrometer lens; 9—fiber tip; 10—optical fiber; 11—pyrometer; 12—CCD camera [36].

Meanwhile, a study by Davis and Shin utilizes a CCD camera(Pulnix TM-1402) in conjunction with a line laser to measure the clad height of the LBPF print [37]. The clad height refers to the height of the deposition of the metal layer. The study created a triangulation-based clad height measurement technique that utilizes planar structured light to directly measure the entire clad cross section's height. See the experimental setup in Figure 11.

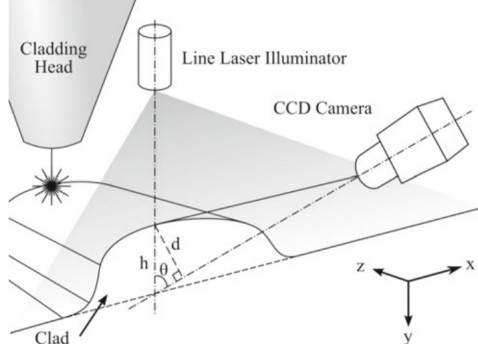

**Figure 11.** Machine Vision system of FDC machine [37].

The clad height *h* appears as distance d in the camera images and is related to the true clad height *h* through.

$$h = \frac{d}{\sin\theta} \tag{3}$$

### 2.2. DSLR (Digital Single Lens Reflex) Cameras

DSLR cameras use a set of mirrors and a pentaprism that allows imaging in the optical viewfinder. This image enters the lens and is captured by the mirror, which is responsible for transferring it to a digital sensor that, in turn, stores the photograph [38].

The cameras are not as sensitive or accurate as CCD cameras primarily because they are not cooled. However, they share some of the advantages of a CCD. They have a linear response over a large portion of their range, and software programs such as AIP4WIN accept their data. They also have their advantages as their fields of view are relatively large [39]. Below Figure 12 shows a schematic of the DSLR camera.

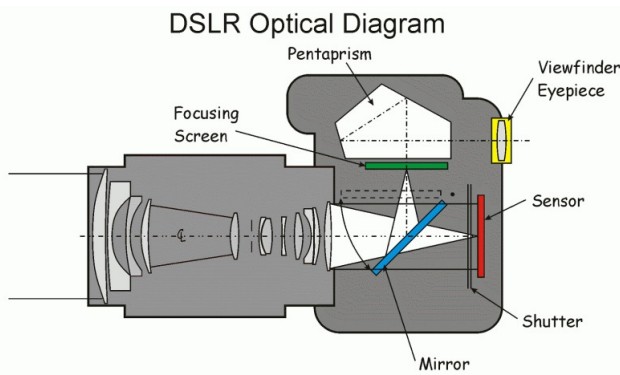

**Figure 12.** Schematic of CCD camera [40].

Petrich et al. had a couple of studies where they used a DSLR CCD camera in conjunction with CT scans to (1) create an anomaly detection algorithm with shorter execution time, (2) create a clustering algorithm that groups anomalous voxels together without pre-defining, (3) make a more robust feature extraction methodology and (4) use data to inform supervised machine learning concepts [41]. The DSLR camera captured eight images of the build platform during different build times.

Timing of image capture was triggered via proximity-sensor monitoring of the recoater blade, with images (1)–(3) captured immediately following the powder recoating operation and photos (4)–(8) captured immediately following the laser fusion step. The machine learning algorithm is further developed with the CT scan images, explicitly using a support vector machine (SVM) in a follow-up study [42]. The errors classified were defined by the machine learning algorithm by similarity of image and were categorized by size and dimension of the discontinuity. A CAD image of the experimental setup is seen in Figure 13.

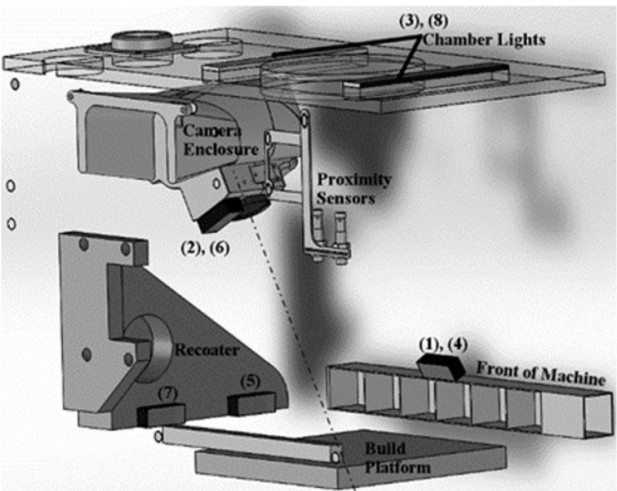

**Figure 13.** Camera systems and light chamber in build chamber [42].

In a study done by Imani et al., they used X-ray computed tomography (XCT) scan data to quantify the effect of laser power (P), hatching space (H), and velocity (V) on the size, frequency, and location of pores. The team also monitored and discriminated the process conditions liable to cause porosity using in-process images captured with a DSLR camera [43]. In Figure 14 you can see the experimental setup used in the study.

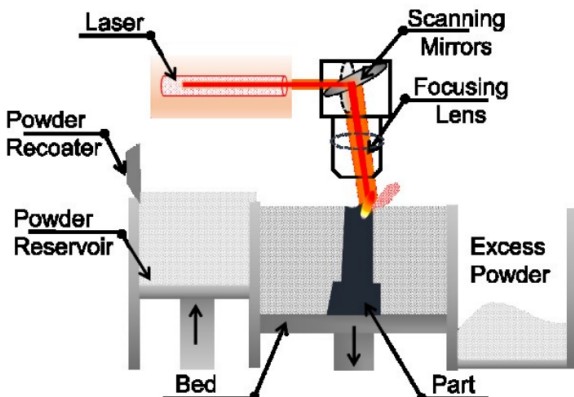

**Figure 14.** Schematic Diagram of the LPBF process [43].

Nine different cylinder prints were built with varying P, H, and V conditions to quantify their effect on the pore spatial distribution frequency, size, and location. Then the images acquired with the DSLR had multifractal and spectral graph theoretic features extracted. In Table 2, we can see the different combinations of process parameters used in creating the cylinders as well as a metric called the Andrew number defined as:

$$E_A = \frac{P}{H \times V} \text{ J/mm}^2 \qquad (4)$$

The Markov Decision Process, an optimal control based on the machine learning algorithm, uses the multifractal components of the work done in this study [44].

**Table 2.** The combination of power (P), hatch spacing (H), scan velocity (V) process conditions used for making the different cylinders [43].

| Process Condition (P, H, V) [W, mm, mm/s] | | $E_A$ [J.mm$^{-2}$] Andrew's Number |
|---|---|---|
| P0, H0, V0 | (340, 0.12, 1250) | 2.27 |
| P −25%, H0, V0 | (250, 0.12, 1250) | 1.70 |
| P −50%, H0, V0 | (170, 0.12, 1250) | 1.13 |
| P0, H+25%, V0 | (170, 0.15, 1250) | 1.81 |
| P0, H+25%, V0 | (170, 0.18, 1250) | 1.51 |
| P0, H0, V0+25% | (170, 0.12, 1562) | 1.81 |
| P0, H0, V0+25% | (170, 0.12, 1875) | 1.51 |

For FFF printing, one study by Nuchitprasitchai et al. builds on research by Pearce [45–47]. The team develops a low-cost, reliable real-time optimal monitoring platform that uses an algorithm for reconstructing 3D images from overlapping 2D intensity measurements for single and double construction [48].

For this paper (Figure 15), there were two camera setups where they either used one camera for 2D reconstruction or two cameras for 3D reconstruction. Each experimental setup uses different algorithms but the same type of camera, printer, and tested objects. Both algorithms used with the two camera setups were effective at detecting the defects of clogged nozzle, loss of filament, or an incomplete project for different 3D object geometries. The difference in shape between STL Image and CameraImage, or the different sizes between STL Image and the 3D reconstruction, informed the error calculation. The printer stopped when these errors exceeded 5%. The two-camera error detection (size error) is more accurate than the one-camera error detection (shape detection); however, the one-camera setup is less computationally intensive. Furthermore, the error details for the double-camera configuration are more comprehensive than the single camera that provided only the total shape error.

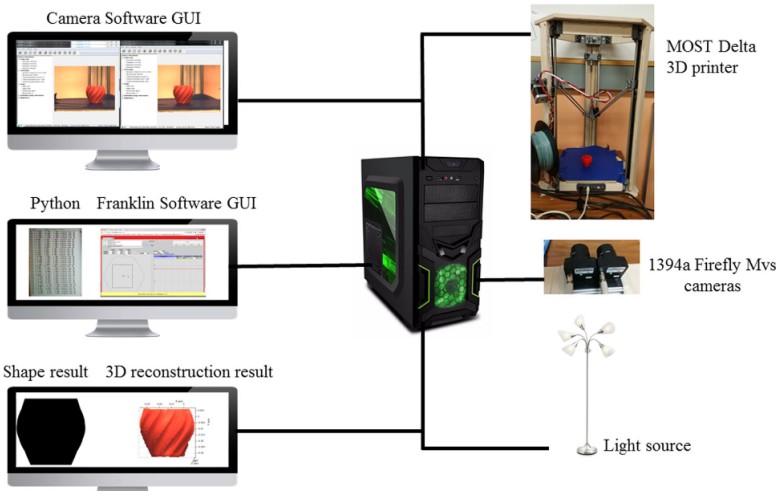

**Figure 15.** Setup of FDM machine, camera, and other sensors of Nuchitprasitchai et al. [48].

Multi-Sensor DSLR Monitoring System

Miao et al. had a system that used multiple sensors to build and compare different models that can predict the distortion and build a system that can prevent distortion based on the prediction model [49]. A customized holder was built to fit an infrared sensor and then mounted onto the print extruder allowing monitoring of the new layer

temperature deposits.The IR sensor information can aid in in-situ prevention of printing when the filament is not extruding or help protect the system by making adjustments in other abnormal situations. Furthermore, the nozzle temperature is monitored by a K-type thermocouple that is mounted on the nozzle through a threaded hole. A Melexis MLX90621 thermal array IR sensor measures the filament temperature. The distortion can be represented by the distance between the highest and lowest points on the deformed part. A Nikon Coolpix P4 camera with 8.1 megapixels and ×3.5 optical zooms measure the distance. See the system architecture in Figures 16 and 17.

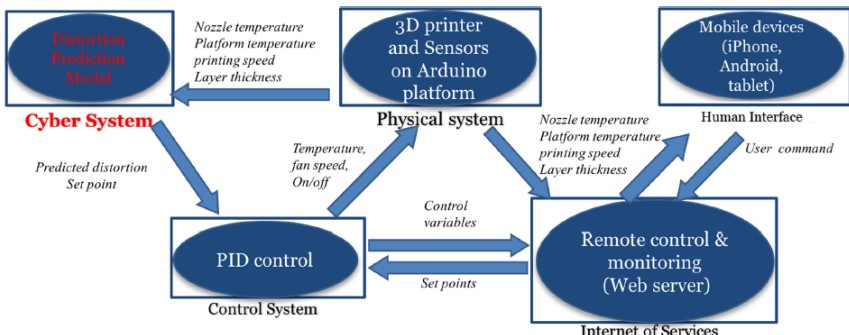

**Figure 16.** Conceptual System Architecture [49].

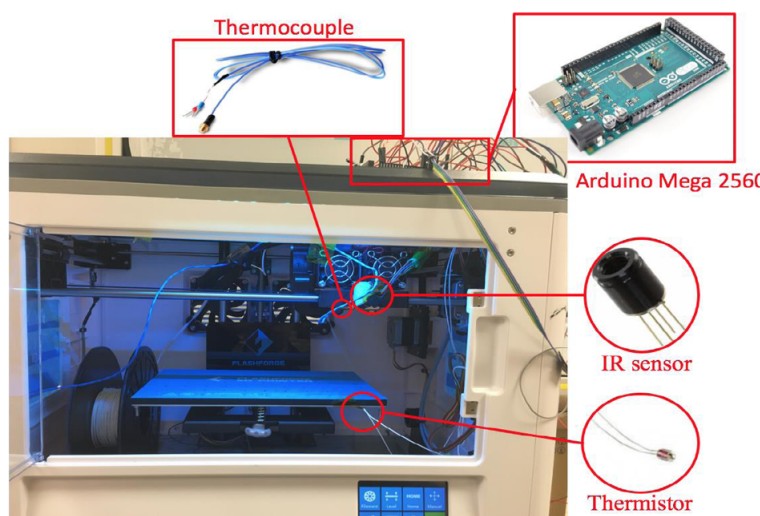

**Figure 17.** Physical System Architecture [49].

The findings of this group were deformation was related to nozzle and platform temperature, the influence of which depended on material and temperature range. For nozzle temperature, in the 200–230 °C range, the deformation of PLA parts will decrease when decreasing the nozzle temperature because the lower nozzle temperature can reduce the temperature gradient. Meanwhile, in the 220–245 °C range, the deformation of ABS parts will decrease when increasing the nozzle temperature. ABS will not melt at 220 °C and will have low liquidity. For the print bed temperature, raising the platform temperature from 110 to 115 °C will increase the distortion possible because the adhesion force decreases when the platform temperature rises. However, this distortion generally plateaus in temperatures higher than 120 °C.

### 2.3. Infrared (IR) Cameras

Infrared imaging cameras are non-contact devices that detect heat or infrared energy. These cameras convert this measure into an electronic signal and then process this information to produce a thermal image [50]. The IR camera detects heat volume to calculate

temperature differences and make clear thermal images in low-light situations. One advantage of using IR cameras is that they are one of the few non-contact ways of detecting heat and can produce images of things in very low-light circumstances. Figure 4 although an image of a CCD camera, is an IR CCD and therefore a helpful schematic to revisit.

Although IR cameras can provide excellent qualitative thermal images, their accuracy is only as good as the accountability of environmental conditions. Calibrating cameras to account for such conditions, therefore, is critical. The radiant energy received from an object will be a function of its temperature, spectral emissivity, reflections from its surroundings, and atmospheric transmission. As radiation passes through the atmosphere, scattering and absorption may attenuate the strength of the signal (Figure 18). Fortunately, IR cameras operate above the wavelength range, which makes these effects negligible. In the regions of minimal disturbance from the atmosphere [51].

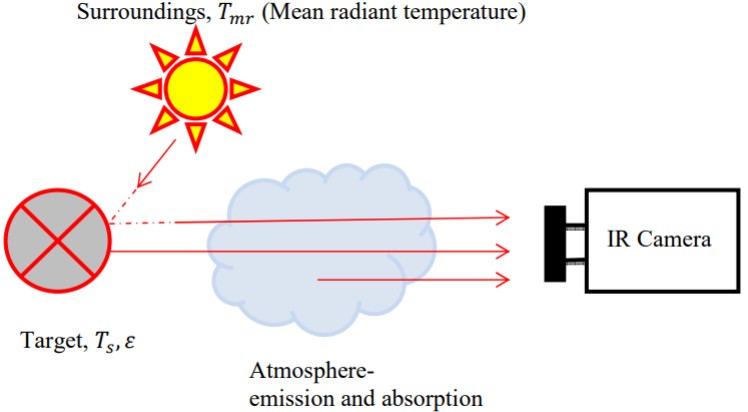

**Figure 18.** Visual of IR Emissivity [51].

Several studies incorporate these IR cameras to monitor the print state of 3D printers. LBPF printers would be the obvious first choice in integrating this technology into their monitoring systems.

Schilp et al. created macro and microscale simulations of the print process. The team measured temperature using a microbolometer IR camera. The typical response time for microbolometer cameras in the order of 8 ms limits the maximum frame rate to approximately 50 Hz. Furthermore, the pixel resolution of 250 μm causes spatial averaging over multiple single scan tracks (width: around 100 μm) [52]. Using the IR camera and modeling software, the team monitored the temperature field of a print on a micro and macro level. The modeling and experiments were created in an effort to see how process parameters like geometry, material, heat input, scanning speed, and ambient influences affected the temperature distribution, melt flow behavior, and wettability of an LPBF print.

Mireles et al. research team modified an Arcam A2 system to include a FLIR SC645 IR camera. The camera saves a thermal history of a build while a print process occurs [53]. After the architecture for the study was set up, the IR camera's monitoring capabilities were evaluated by intentionally seeding porosity defects into a part assembly from 100 μm to 200 μm. See the group set up in Figure 19. Different shapes were utilized, including spheres, triangular prisms, cylinders, and cubes. In detecting the defects, the study first uses IR image analysis, which failed to find any flaws more minor than 600 μm [54]. The defects were then X-ray CT scanned and then underwent a destructive analysis to compare all forms of part analysis. The process parameter used to correct defects in situ is re-scanning, where the affected area is re-melted. For post-processing, a process called hot isostatic pressing (HIP) was used to better the part's material properties.

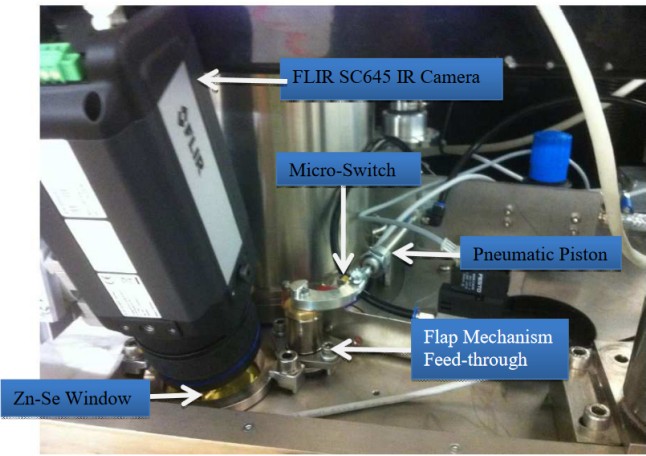

**Figure 19.** Top of ARCAM A2 Build Chamber with new component location and IR camera [51].

FDM printing also uses thermal imaging to monitor defects in a print. A series of studies by Costa et al. examines the contribution of heat transfer of various thermal phenomena during manufacturing [55]. The group used a FLIR ThermaCAM SC640 IR camera to record the surface temperature of the filament. Figure 20 shows a picture of the setup. The main aspect being tested and monitored in this study was the physical contact between filament segments during the progressive build-up of the 3D structure, which is tested by an adhesion test [56]. The parameters that were altered in the model were processing conditions (extrusion temperature, environment temperature, extrusion velocity, filament diameter etc.), material properties and depsoition sequence.

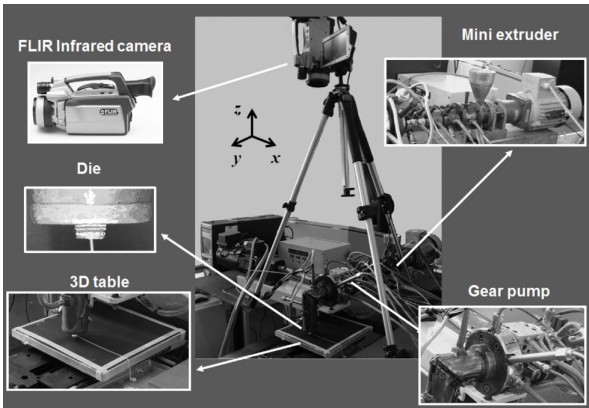

**Figure 20.** Experimental setup for Costa et al. [56].

Ferraris et al. utilize an Optris PI640 IR camera on a commercial Prusa MK3 to capture temporal temperature variations in the x-z plane. The study's goal was to capture spatial and temporal temperature variations while printing an FDM/FFF part, which would enable a combined experimental and numerical approach for in-process monitoring [57]. See the groups experimental setup in Figure 21. The parameters that were varied was the time that the printing proccess was monitored changing the boundary conditions of both the inter-layer and intra-layer time. The second parameter that was changed was the location of where the IR camera was monitoring varying between the 3rd, 20th, and 38th layer and varying x positions of 2, 9, and 16 mm.

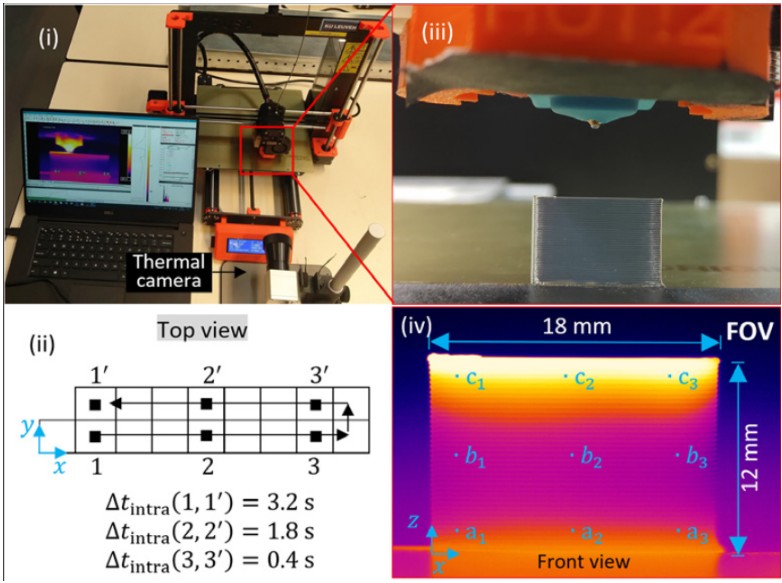

**Figure 21.** (**i**) Picture of the thermal set-up for in-process monitor of the FFF process; (**ii**) top view of the sample geometry, deposition sequence and intra-layer time ( $\Delta t_{intra}$ ) for elements located on given x-y planes; (**iii**) picture of a realised sample and (**iv**) thermogram of the printed part. Given x-z locations are highlighted [57].

Li et al. propose a framework to integrate physically-based and data-driven approaches for component scale and layer-to-layer thermal field prediction.

As seen in Figure 22, the framework captures the underlying physical printing processes. A physically-based model is built and simulated using 3D transient finite element analysis (FEA). Sampling in the design space of process inputs, historical FEA simulation results in a surrogate model [58]. The resultant model links model inputs to model outputs. The actual experimental measurements were compared with the model framework through Bayesian calibration which allowed identification of the unknown model parameters and model discrepancy. This calibrated the model in a non-parametric way. Due to the model being trained and updated with the experimental data collected in-situ, it can capture the physical processes in FDM as well as correct some model discrepancies that are associated with the imperfect understanding of the underlying physics.The model is cross-validated with a new process setting and new geometric designs. The process settings that were used in the model include layer thickness, printing speed, nozzle temperature, layer index, printing pattern direction, and neghborhood time difference (NTD). Additionally there were two calibration parameters of the heat convection coefficient and latent heat of fusion. A FLIR A6555sc IR camera did the thermal imaging.

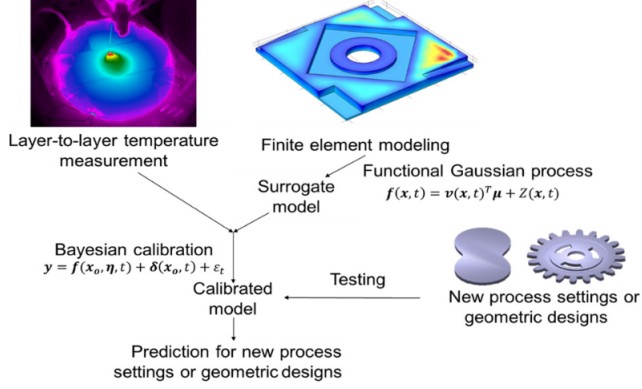

**Figure 22.** Proposed Framework [58].

Hu et al. utilize a support vector machine with thermal imaging to diagnose FDM printing faults caused by the variation of temperature fields. See the architecture setup in Figure 23 [59]. Support vector machine (SVM) is a pattern recognition method based on statistical learning theory. Due to its good generalization ability, especially towards serious nonlinear problems, SVM often has uses as a diagnostic tool for FDM parts. ABS plastic parts are built on a Makerbot Creator with a VarioCAM®hr-HS as the thermal imager. A process where the nozzle temperature gradually decreases each layer printed various factors. The temperature gradation was from 220 °C to 180 °C. The variable temperature prints pair against a control where the nozzle temperature doesn't change. The different defects are classified with data analysis. The print defects were broken up into two stages. In stage one the defects are simply normal printing, which is also regarded as on of FDM printing states for modeling convenience, and abnormal printing. Meanwhile in stage two the printing defects include insufficient filling warping serous fault printing and print failure.

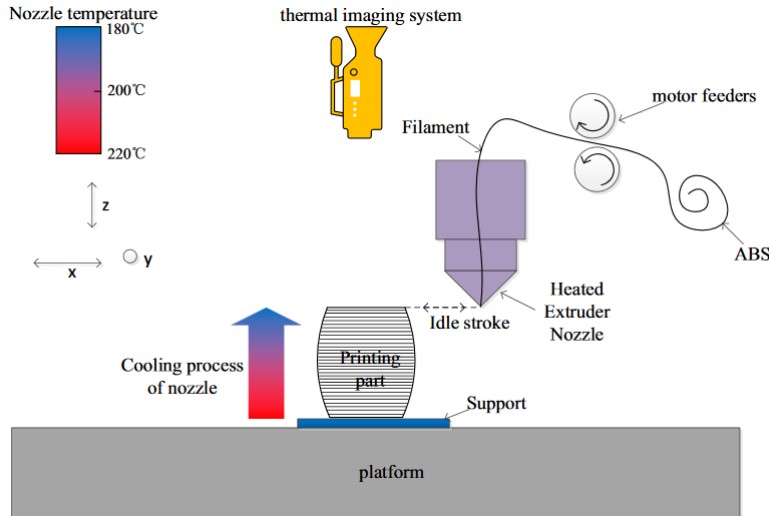

**Figure 23.** Printing Process Diagram [59].

*2.4. USB Cameras*

USB cameras have become synonymous with the name web camera; however, with the development of these cameras, they have uses outside of video conferencing. USB cameras connect mainly through a USB port where the video feeds to the computer. A software application allows viewing of the pictures and transfers them to the internet [60]. The main downside to USB cameras is that a PC has to be switched on at the camera's location to have images taken.

Khandpur et al. use a USB camera by Svpro that comes equipped with a Sony IMX322 sensor. The camera was a cost-effective choice with an effective enough resolution. The camera costs around 85 euros, can focus up to 5 mm, and has a resolution of up to 2 Megapixels [7]. The experimental setup can be seen in Figure 24.

The camera was positioned into an A4v3 3D printer by 3ntr, capturing a top-down view of the print bed. The group does not specify a type of error but compares the top-down image of a print layer and compares this to the code. With image processing and a user interface designed in the Matlab App designer, a flag arises if a defect is detected, and the program shows the fault area. The process parameters that were controlled in the system were the layer height, nozzle temperature, and the bed temperature. See the Matlab interface in Figure 25.

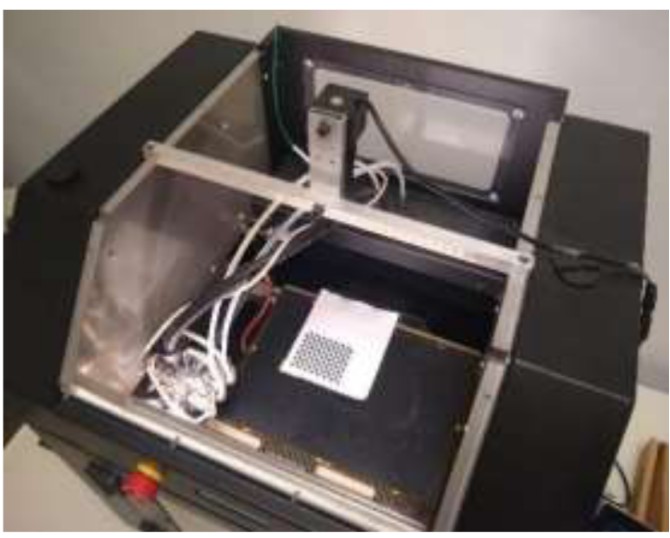

**Figure 24.** Printer setup with the Svpro camera and checkerboard pattern [7].

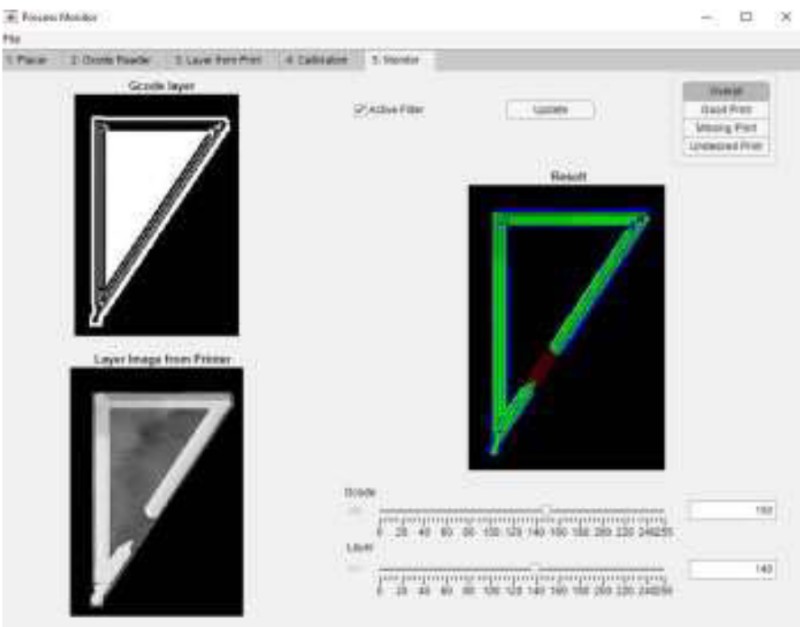

**Figure 25.** Matlab Interface [7].

The process of image processing to find areas of error can be seen in Figure 26.

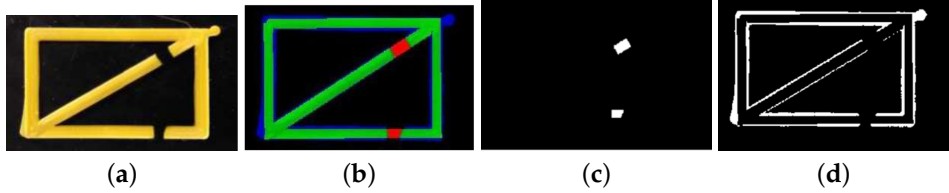

**Figure 26.** (**a**) Printed sample with artificial defects (**b**) Results of process monitoring (**c**) Area corresponding to artificial defect (**d**) Edge detectopm.

Multi-Sensor USB Camera System

Moretti et al. worked to incorporate multi-sensor data fusion technology into AM manufacturing to create a "smart" machine that could monitor the manufacturing process and analyze part quality [61]. The multi-sensor system includes encoders on the XYZ axis, a J-type thermocouple for the hot end temperature, and a Logitech C170 USB camera to

image the print bed. An Open Electronics 3DRag printer is the test printer used. See the multi-sensor printer setup in Figure 27.

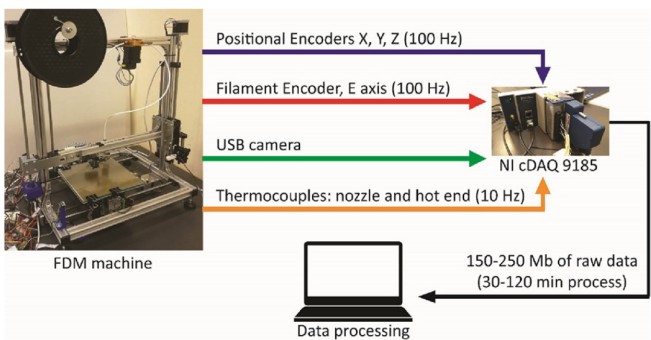

**Figure 27.** Schematics of the sensor data streams captured by the NI cDAQ [61].

Although some print tests were done that changed the infill density of parts and geometry of parts, the main focus was on the architecture's data-fusion problem. It looks at how the data signals take and how accurately they measure reality.

Gao et al. also created a multi-sensor monitoring sensor to make a defense mechanism against cyber-physical attacks on 3D printers, looking at both the kinetic and thermodynamic prospects [62]. The group utilizes an Ultimater 2 Go Desktop FDM 3D printer for the machine they will monitor. The attached sensors include an inertial measurement unit (IMU), an accelerometer, and a Logitech C960 webcam using both the image and sound data for monitoring. The team monitors the infill path, printing speed, layer thickness, and fan speed estimation. In Figure 28, the process of checking for attack and correcting for is added into a general 3D printing process.

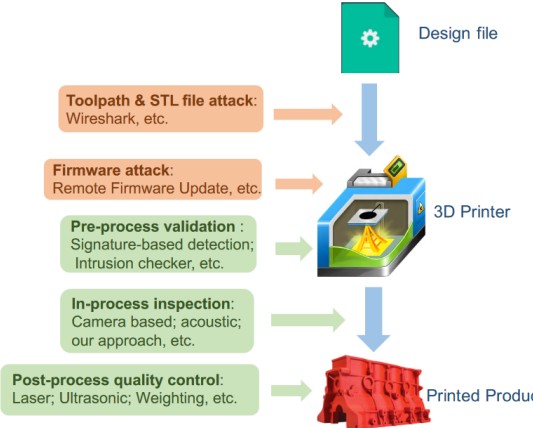

**Figure 28.** Taxonomy of attacks and defenses in additive manufacturing process chain [62].

### 2.5. Miscellaneous Cameras

Multiple other types of cameras and imagers are in AM process monitoring systems. For the sake of not having a section for every camera iteration, some notable studies that don't fall under the above categories are in the miscellaneous camera category. These cameras tend to be stock cameras you can get at a lower price point or particular cameras to focus directly on a given issue.

To begin, Baumann and Roller utilize a Playstation EyeCam with an OmniVision chip to image the printing process of a Makerbot Replicator 2X [63]. This camera is very similar to how a USB camera would work but is slightly different as it is formatted to work for Playstation and therefore doesn't have a PC interface [64]. In Figure 29, you can see the EyeCam, which looks very similar to a standard webcam.

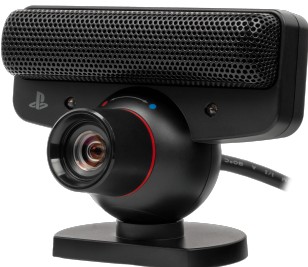

**Figure 29.** Image of Sony Ps3 EyeCam [65].

With the Omnivision chip, some extra filters and sensors might not be evident in a general webcam [66]. The EyeCam is placed with a side-view orientation; therefore, no complete view of the print bed exists in the study. The type of errors the study attempted to catch included detachment, missing material flow, deformed objects, surface errors, and general deviation from the model. Optical markers are placed on the horizontal ground line of the print bed as well as the print head to minimize computational effort. These markers allow auto-cropping of the video image. External color thresholding was used for pre-processing, while blob detection was a large part of the in-situ error detection. In the test prints, the process parameter that was controlled was the filament flow. The team would purposely cut the filament flow to seed errors into a print.

Ceruti et al. take an exciting approach to error detection by utilizing augmented reality (AR). They define virtual reality as the user interacting with a completely virtual scene, while augmented reality always has some contact with the real world [67]. Below in Figure 30 is the apparatus of the AR glasses camera and 3D printer.

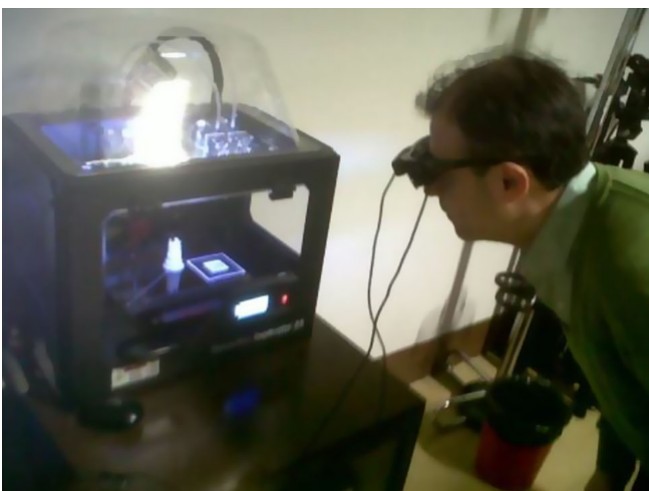

**Figure 30.** Printing machine with camera (top) or Wuzix glasses (down) [67].

The proccess used in the AR procedure is as follows.

1. Image acquisition: using an internal camera (digital camera connected to PC) or external glasses fit by the experimenter
2. Calibration: evaluate internal parameters of camera to correct for image distortion
3. Tracking: defines the position in space and orientation of camera with respect to fixed references
4. Registration: synchronizing virtual and real world image
5. Display: virtual object added on top of real word image.

After applying the AR program to the printing environment, the lab group utilizes a speeded-up robust features (SURF) algorithm. The SURF algorithm is a scale and rotation invariant procedure to detect interest points in images like corners, edges, points with

sharp changes, etc. After splitting the part into distinct features, the actual print can be compared to a CAD augmented reality image to locate particular errors where the image and real-world print features are significantly skewed. The group created one case study where the print was stretched 5 mm more on the wings of the print compared to original CAD to test error detecting capabilities. However, further development regarding error correction and more detailed print anomalies requires research as the paper was mainly a proof of concept.

Multi-Sensor Miscellaneous Camera System

Rao et al. used multi-sensor fusion to study the states where print failure would occur [8]. The sensors utilized were accelerometers, thermocouples, a non-contact IR sensor, and a borescope for live visual feed. A borescope is an optical tool to view areas that would otherwise not be visible. The camera inserts into the evaluated item without destroying the object of interest [68]. See an image of a borescope and its components in Figure 31 below.

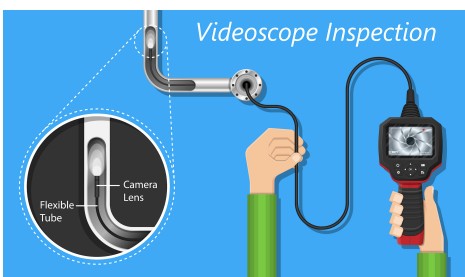

**Figure 31.** Borescope and corresponding components [69].

The group uses a USB borescope by Supereyes [70] that has a 13 mm diameter and built-in measuring functions. The borescope works similar to a webcam presenting a live feed of the print in its entirety. Using a borescope allowed more up-close imaging of the extrusion process than other general camera types. The group analyzes three process variables: extruder temperature, feed rate to flow rate ratio, and layer thickness and build quality (which, in this paper, is the arithmetic average surface roughness of the print). The group used a Dirichlet process and evidence theoretic (DP+ ET) to establish an effective sensor fusion technique and a process fault classification approach. Sensor data acquired under various process conditions show how sensor signal patterns were associated with different process states and the evolution of build failures. These signal patterns helped to identify the process setting that led to building failures and the physical root cause of said failures. See the experimental setup in Figure 32.

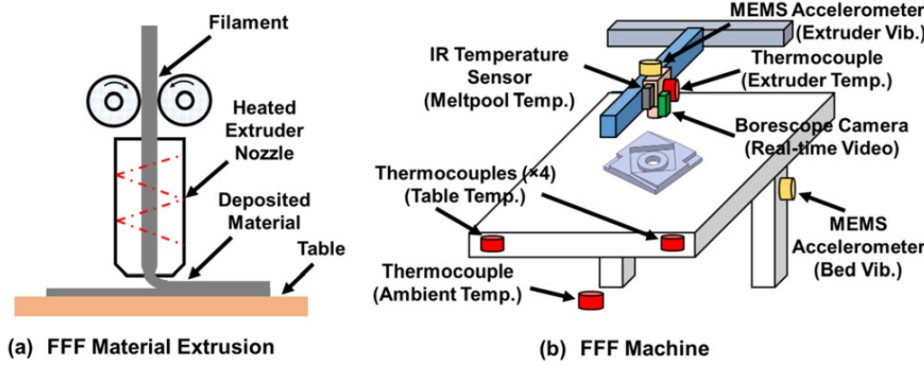

**Figure 32.** (**a**) Schematic of the FFF process. (**b**) Schematic of the FFF setup instrumented with multiple in situ sensors used in this work for measuring process conditions in real-time [8].

## 3. Discussion

One of the main distinctions of the architectures is less about the specific camera and more about the number of sensors used. The architecture that only utilizes one camera can detect one-dimension errors. The layer height is the leading standard parameter monitored for experiments using only a single camera for sensing capabilities.

Once we get into multi-sensing territory, the errors that are capable of detection become fuzzier. Generally speaking, wanting to detect the location of specific 2D errors requires two visualization methods. Often this visualization comes down to a camera that visualizes the side of the print and another that gets a top-down image. In the studies in which this isn't present (for instance [67], there is generally CAD assistance, and the error verification is not precise).

Regarding mapping process failure to specific errors, sensors monitor the internal inter-workings of the printer. IR cameras seem to be the most utilized in these cases in conjunction with other sensors. In being able to map the heat of the bed/print visually and having thermocouples, accelerometers, etc., on the parts, we can have redundant data that is compared to each other and synthesized to give a fuller understanding of what is happening during a vibrant print.

Cost is essential in choosing desired architecture outside of talk of error. Generally speaking, LPBF AM monitoring will be expensive anyway, as the printers are not cheap. However, webcams and non-specific miscellaneous cameras will be your most affordable option when discussing additional optical sensors and FDM architecture. CCD cameras and DSLR cameras, on the other hand, will have a general proportional cost-to-quality ratio. Other costs to account for are what you are getting with your purchase. Many miscellaneous cameras have in-built sensors or software that monitor different aspects of the print other than just visualization of the print bed. In contrast, if you get a stock camera, you will have mainly only visualization features.

A final note, with analysis, is that although the lighting architecture isn't mentioned much in this paper, this is a crucial part of the monitoring architecture of a print. There are often trade-offs in the angle at which the camera should be positioned and how to illuminate the printing area to get valid data. The user who plans on creating their own AM monitoring system will have to keep the type of printer they will buy, the size of camera they need, and how many sensors they can realistically fit on the apparatus.

Further discussion on errors and architecture specific to each printer type is provided below. These trends are summarized in Tables 3 and 4.

### 3.1. FDM

CCD cameras tend to be the standard optical monitors for the one-dimensional errors seen in FDM printers. Although other cameras can be used to find one-dimensional errors, if that is the case, these errors(generally infill layer and layer height) are a set of many errors that are trying to be characterized. For this one-dimensional error, the process parameter that is the controlled variable is usually the flow rate. However, Shen et al. [35] monitored several other reasons for the effects found in their study to occur, including nozzle temperature, layer thickness, ambient parameters and the aforementioned flow rate.

For FDM printing, 2D errors are not as defined as most one-dimensional errors and usually are spotted using some machine learning algorithm and seeding the algorithm with author-defined errors. The 2D errors are created based on different texturization methods. Some algorithms that detect error include artificial neural networks, support vector machines, nearest neighbor, Dirichlet Process, and more.

FDM printers are much more likely to be used and experimented on because of their low price, especially for low-end printers. This is promising for developing process monitoring for this style of AM. However, in the same vein, any pre-built monitoring software is rare for this printer style; therefore, much experimentation and trial and error are needed for even your most basic set-up.

**Table 3.** Summary of FDM studies.

| Author | Optical Method | Process Parameter | Multi-Sensor | Error |
|---|---|---|---|---|
| Cheng and Jafari [32] | CCD | Roller Speed | No | road width (underfill or overfill) |
| Shen et al. [35] | CCD | Nozzle temperature, environment temperature, nozzle diameter, layer thickness, feed velocity | No | Tranverse defect, longitudinal defect, localized defect |
| Nuchitpras- itchai et al. [48] | DSLR | clogged nozzle, loss of filament, incomplete project | No, one and dual camera cases | geometric difference between CAD and capture of print |
| Miao et al. [49] | DSLR | Nozzle and platform temperature | IR Sensor, K-type thermocouple | distortion |
| Costa et al. [55,56] | IR | processing conditions,material properties, deposition sequence | No | cross-section deformation and layer adhesion |
| Ferraris et al. [57] | IR | Location and time of section monitored | No | temperature variations |
| Li et al. [58] | IR | layer thickness, printing speed, nozzle temperature, layer index, printing pattern direction, neighborhood time difference | No | thermal field prediction |
| Hu et al. [59] | IR | nozzle temperature | No | Insufficient filling, warping, serious fault printing, print failure |
| Khandpur et al. [7] | USB | Layer Height, Nozzle temperature, bed temperature | No | top down geometrical difference of print and GCode |
| Moretti et al. [61] | USB | infill and part geometry | XYZ Encoders, J-Thermo- couple | Data Fusion |
| Gao et al. [62] | USB | infill path, printing speed, layer thickness | IMU and accelerometer | verifying print parameters |
| Baumann and Roller [63] | Play station EyeCam | filament flow | No | detachment, missing material flow, deformed objects, surface errors, geographical deviation from model |
| Rao et al. [8] | Bore- scope | Extruder temp, feed rate, flow rate ratio, layer thickness | accele- rometer, thermocouples, IR sensor | normal, abnormal, and failure |
| Ceruti et al. [67] | AR glasses | None | No | Geometric differences |

*3.2. LPBF*

For the LBPF printers, the one-dimension defect monitored is the powder bed height. Whereas the general 2D parameters monitored are porosity and how said porosity affects the print as a whole. Often this is achieved with a combination of an in-built CT scanner in the printer and an additional CCD camera.

Because LBPF printing parameters are tied together more directly using the energy density function, any specific impact of process parameters on the print build is usually coupled with at least one other parameter. For this reason, most studies dealing with errors larger than the one-dimensional errors tended to be either just monitoring studies where a couple of process parameters were changed through a set range and different maps of the the effects generated. Or a machine learning algorithm used where the AI auto generated specific errors depending on similarity of error visually.

As mentioned in the FDM chapter, LPBF printers tend to be at a pricier range and therefore less likely to have success being used in a wide array of research as a cheaper printer would be. However, due to certain in-built sensor like CT scanning already being incorporated in many printers, optical monitoring can be attempted without any additional modifications to the build.

**Table 4.** Summary of LPBF studies.

| Author | Optical Method | Process Parameter | Multi-Sensor | Error |
|---|---|---|---|---|
| Kleszczynski et al. [21,28] | CCD | energy density,scanning speed, overhanging angle | No | balling formation, super elevation, support connects |
| Zeinali et al. [31] | CCD | flow rate, powder density, scan speed, jet diameter, laser beam diameter, nozzle angle, laser angle | No | Clad Height |
| Doubenskaia et al. [36] | CCD | hatch distance, powder layer thickness | Pyrometer | Monitoring how process parameters affect one another |
| Davis and Shin [37] | CCD | Clad-height | Line-laser | Validate measurement technique |
| Peitrich et al. [41,42] | DSLR | Machine Learning model | No, CT post scan | Anomoly Detection of Voxels |
| Imani et al. [43] | DSLR | laser power, hatching space, velocity | No, post X-ray CT scan | porosity |
| Schilp et al. [52] | IR | Scanning Speed | No | temperature distribution, melt flow behavior, wettability |
| Mireles et al. [53,54] | IR | re-scanning | No, post X-ray CT scan | porosity |

## 4. Conclusions

This literature review was meant to look into the experimental setups of AM monitoring systems from an optical point of view. The two most prominent in literature AM processes were focused on, were FMD and LPBF printing. The review looked at how different cameras and architectural styles lend themselves to discovering and possibly altering different errors in the prints. The review found a couple of trends.

1. Regarding state-of-the-art AM monitoring, there seems to be a significant gap in knowledge in terms of texture analysis as well as mapping errors to a given cause. There appears to be a growing amount of research on how a product looks with a given parameter change. In the same vein, an interest in studies that monitor more closely all of the process signals during a print. However, there needs to be a growing knowledge of how specific mechanical failures of a 3D printer affect a print and, more specifically, how to fix or compensate for these mechanical errors. The one-dimensional errors have a more considerable progression of understanding in this regard. However, research on textural mistakes and their root causes is still early.

2. Another section of infancy is in-situ error correction. Many of the methods in the papers reviewed required post-print analysis, which would make any form of correction impossible. Similar to finding root causes of errors, in-situ detection and correction of one-dimensional errors has been attempted, although very much in its infancy stages. Any textural in-situ error correction has either been limited in scope or non-existent.

With this in mind, the most promising direction to help further the field of AM monitoring would be creating a multi-sensing architecture with sensors that could give back real-time results. The system should analyze data to inform the printer what it should do next. An architecture like this would need at least three sensors, 2 of which would be optical and the third most likely thermal (if one of the two optical sensors is not IR). Further recommendations include XYZ encoders and an encoder on the filament motor (regarding FDM) to monitor the difference between the desired location of the print head and the actual.

**Author Contributions:** Conceptualization, B.W.; methodology, B.W.; software, B.W.; validation, C.M.J.; formal analysis, B.W.; investigation, B.W.; resources, C.M.J.; data curation, B.W.; writing—original draft preparation, B.W.; writing—review and editing, B.W.; visualization, B.W.; supervision, B.W.; project administration, B.W.; funding acquisition, C.M.J. All authors have read and agreed to the published version of the manuscript.

**Funding:** This research was funded by the Title III Graduate Engineering Program.

**Data Availability Statement:** No new data were created or analyzed in this study. Data sharing is not applicable to this article.

**Conflicts of Interest:** The authors declare no conflict of interest.

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
