# Peer review of "Optical Methods of Error Detection in Additive Manufacturing: A Literature Review"

_jmmp, doi:10.3390/jmmp7030080_

Round 1

Reviewer 1 Report

This paper tries to give a review on Optical Methods of Error Detection in Additive Manufacturing. However, I am afraid it is not thorough and has no scientific contributions.

1. “This paper looks through the literature on two AM processes: fused deposition modeling (FDM) and laser bed powder fusion (LBPF ) printers”, why only these two AM techniques?

2. This paper only lists some applications of AM using CCD, DSLR, Infrared (IR) Cameras, USB Cameras, etc.; I am afraid there is no scientific contribution seen in this paper. No deep analysis, no future perspectives, just list some cases.

Author Response

1) FDM printing is one of the primary forms of additive manufacturing, and thus a paper mainly focusing on this style of printing is both helpful and necessary. Furthermore, because of specific in-built sensors, a wealth of knowledge of monitoring LPBF printers already exists in the literature and is generally cited even in non-SLA style reviews.

2) Studies main contribution is to help to see how the architecture of monitoring systems limit ability to monitor certain errors and vice versa.

Reviewer 2 Report

The current manuscript presents a literature review on the optical monitoring systems of error detection in additive manufacturing. The presented data are comprehensive and valuable. However, some issues should be considered as follows:

- The introduction section should include a brief of the efforts achieved in the literature studies to optimize the process parameters and parts' quality for both FDM and PBF techniques. Followed by the impact and the expected improvement of the quality of the fabricated parts using optical monitoring systems.   

as an example, the following studies can add more data regarding the above point;

* Maamoun, A.H.; Xue, Y.F.; Elbestawi, M.A.; Veldhuis, S.C. Effect of Selective Laser Melting Process Parameters on the Quality of Al Alloy Parts: Powder Characterization, Density, Surface Roughness, and Dimensional Accuracy. Materials 201811, 2343. https://doi.org/10.3390/ma11122343

DebRoy T, Wei HL, Zuback JS, Mukherjee T, Elmer JW, Milewski JO, Beese AM, Wilson-Heid AD, De A, Zhang W. Additive manufacturing of metallic components–process, structure and properties. Progress in Materials Science. 2018 Mar 1;92:112-224.

- It is recommended to add 2 subsections to the discussion to display the data for the optical monitoring of FDM and PBF techniques separately. 

- For the conclusion section, it is recommended to use a bullet points style to focus on the main conclusions, recommendations, and future perspective.   

looking forward to addressing the above recommendation in the revised manuscript.  

Author Response

1) Added additional information on how process parameters affected printing defects

2) Separated Discussion section into general observations, FDM, and then LPBF

3) Put most of my summarized points in conclusion in numbered list and after put recommendations for future

Author Response

1) Done

2) Tried to add more passive voice

3)There are mainly literature reviews on SLA printers and not as many on FDM. Most papers focus on methods for finding specific errors on the algorithm side but do not focus as much on describing how limitations of architecture can affect the ability of a system to find an error

4) Added more information on detection techniques in process parameter section of the introduction

5) Improved all figures other than figures 23 and 24 as the original figures in the paper also have poor resolution

6) Updated FDM conclusion table

7) I'm mainly focusing on hardware limitations so I can help in providing a guide on what camera setup/sensors would be helpful to use for a given error. Image processing is more a software/machine learning concept that is important for building a monitoring system but outside the review's scope

8) Most of the papers cited are multi-sensor systems. I separated the multi-sensor papers by what camera was used to visualize the printing process

9) Fixed equations

11) Mentioned future prospects/trend in discussions in conclusion

Round 2

Reviewer 1 Report

This paper has been improved. The reviewer has the following minor concerns. 1. "AM’s printing process typically stacks material layer-by-layer to construct three-dimensional (3D) products.", for this statement, suggest to include some references, such as "A survey of additive manufacturing reviews, https://accscience.com/journal/MSAM/1/4/10.18063/msam.v1i4.21"; "Additive manufacturing technologies, Springer book, 2021.".

Author Response

Updated

Reviewer 2 Report

The revised manuscript is improved. However, as a minor point; there is no need to numbering the following subsections (2.1.1, 2.2.1, 2.4.1, 2.5.1).

Author Response

Updated